# Dependent Multinomial Models Made Easy: Stick Breaking with the Pólya-Gamma Augmentation

**Scott W. Linderman**[*]
Harvard University
Cambridge, MA 02138
swl@seas.harvard.edu

**Matthew J. Johnson**[*]
Harvard University
Cambridge, MA 02138
mattjj@csail.mit.edu

**Ryan P. Adams**
Twitter & Harvard University
Cambridge, MA 02138
rpa@seas.harvard.edu

## Abstract

Many practical modeling problems involve discrete data that are best represented as draws from multinomial or categorical distributions. For example, nucleotides in a DNA sequence, children's names in a given state and year, and text documents are all commonly modeled with multinomial distributions. In all of these cases, we expect some form of dependency between the draws: the nucleotide at one position in the DNA strand may depend on the preceding nucleotides, children's names are highly correlated from year to year, and topics in text may be correlated and dynamic. These dependencies are not naturally captured by the typical Dirichlet-multinomial formulation. Here, we leverage a logistic stick-breaking representation and recent innovations in Pólya-gamma augmentation to reformulate the multinomial distribution in terms of latent variables with jointly Gaussian likelihoods, enabling us to take advantage of a host of Bayesian inference techniques for Gaussian models with minimal overhead.

## 1   Introduction

It is often desirable to model discrete data in terms of continuous latent structure. In applications involving text corpora, discrete-valued time series, or polling and purchasing decisions, we may want to learn correlations or spatiotemporal dynamics and leverage these structures to improve inferences and predictions. However, adding these continuous latent dependence structures often comes at the cost of significantly complicating inference: such models may require specialized, one-off inference algorithms, such as a non-conjugate variational optimization, or they may only admit very general inference tools like particle MCMC [1] or elliptical slice sampling [2], which can be inefficient and difficult to scale. Developing, extending, and applying these models has remained a challenge.

In this paper we aim to provide a class of such models that are easy and efficient. We develop models for categorical and multinomial data in which dependencies among the multinomial parameters are modeled via latent Gaussian distributions or Gaussian processes, and we show that this flexible class of models admits a simple auxiliary variable method that makes inference easy, fast, and modular. This construction not only makes these models simple to develop and apply, but also allows the resulting inference methods to use off-the-shelf algorithms and software for Gaussian processes and linear Gaussian dynamical systems.

The paper is organized as follows. After providing background material and defining our general models and inference methods, we demonstrate the utility of this class of models by applying it to three domains as case studies. First, we develop a correlated topic model for text corpora. Second, we study an application to modeling the spatial and temporal patterns in birth names given only sparse data. Finally, we provide a new continuous state-space model for discrete-valued sequences,

---

[*]These authors contributed equally.

including text and human DNA. In each case, given our model construction and auxiliary variable method, inference algorithms are easy to develop and very effective in experiments.

Code to use these models, write new models that leverage these inference methods, and reproduce the figures in this paper is available at `github.com/HIPS/pgmult`.

## 2 Modeling correlations in multinomial parameters

In this section, we discuss an auxiliary variable scheme that allows multinomial observations to appear as Gaussian likelihoods within a larger probabilistic model. The key trick discussed in the proceeding sections is to introduce Pólya-gamma random variables into the joint distribution over data and parameters in such a way that the resulting marginal leaves the original model intact.

The integral identity underlying the Pólya-gamma augmentation scheme [3] is

$$\frac{(e^\psi)^a}{(1+e^\psi)^b} = 2^{-b} e^{\kappa\psi} \int_0^\infty e^{-\omega\psi^2/2} p(\omega \,|\, b, 0) \, \mathrm{d}\omega, \tag{1}$$

where $\kappa = a - b/2$ and $p(\omega \,|\, b, 0)$ is the density of the Pólya-gamma distribution $\mathrm{PG}(b, 0)$, which does not depend on $\psi$. Consider a likelihood function of the form

$$p(x \,|\, \psi) = c(x) \frac{(e^\psi)^{a(x)}}{(1+e^\psi)^{b(x)}} \tag{2}$$

for some functions $a$, $b$, and $c$. Such likelihoods arise, e.g., in logistic regression and in binomial and negative binomial regression [3]. Using (1) along with a prior $p(\psi)$, we can write the joint density of $(\psi, x)$ as

$$p(\psi, x) = p(\psi) \, c(x) \frac{(e^\psi)^{a(x)}}{(1+e^\psi)^{b(x)}} = \int_0^\infty p(\psi) \, c(x) \, 2^{-b(x)} e^{\kappa(x)\psi} e^{-\omega\psi^2/2} p(\omega \,|\, b(x), 0) \, \mathrm{d}\omega. \tag{3}$$

The integrand of (3) defines a joint density on $(\psi, x, \omega)$ which admits $p(\psi, x)$ as a marginal density. Conditioned on these auxiliary variables $\omega$, we have

$$p(\psi \,|\, x, \omega) \propto p(\psi) e^{\kappa(x)\psi} e^{-\omega\psi^2/2} \tag{4}$$

which is Gaussian when $p(\psi)$ is Gaussian. Furthermore, by the exponential tilting property of the Pólya-gamma distribution, we have $\omega \,|\, \psi, x \sim \mathrm{PG}(b(x), \psi)$. Thus the identity (1) gives rise to a conditionally conjugate augmentation scheme for Gaussian priors and likelihoods of the form (2).

This augmentation scheme has been used to develop Gibbs sampling and variational inference algorithms for Bernoulli, binomial [3], and negative binomial [4] regression models with logit link functions, and to the multinomial distribution with a multi-class logistic link function [3, 5].

The multi-class logistic "softmax" function, $\boldsymbol{\pi}_{\mathsf{LN}}(\boldsymbol{\psi})$, maps a real-valued vector $\boldsymbol{\psi} \in \mathbb{R}^K$ to a probability vector $\boldsymbol{\pi} \in [0, 1]^K$ by setting $\pi_k = e^{\psi_k} / \sum_{j=1}^K e^{\psi_j}$. It is commonly used in multi-class regression [6] and correlated topic modeling [7]. Correlated multinomial parameters can be modeled with a Gaussian prior on the vector $\boldsymbol{\psi}$, though the resulting models are not conjugate. The Pólya-gamma augmentation can be applied to such models [3, 5], but it only provides single-site Gibbs updating of $\boldsymbol{\psi}$. This paper develops a joint augmentation in the sense that, given the auxiliary variables, the entire vector $\boldsymbol{\psi}$ is resampled as a block in a single Gibbs update.

### 2.1 A new Pólya-gamma augmentation for the multinomial distribution

First, rewrite the $K$-dimensional multinomial recursively in terms of $K - 1$ binomial densities:

$$\mathrm{Mult}(\boldsymbol{x} \,|\, N, \boldsymbol{\pi}) = \prod_{k=1}^{K-1} \mathrm{Bin}(x_k \,|\, N_k, \widetilde{\pi}_k), \tag{5}$$

$$N_k = N - \sum_{j<k} x_j, \quad \widetilde{\pi}_k = \frac{\pi_k}{1 - \sum_{j<k} \pi_j}, \quad k = 2, 3, \ldots, K, \tag{6}$$

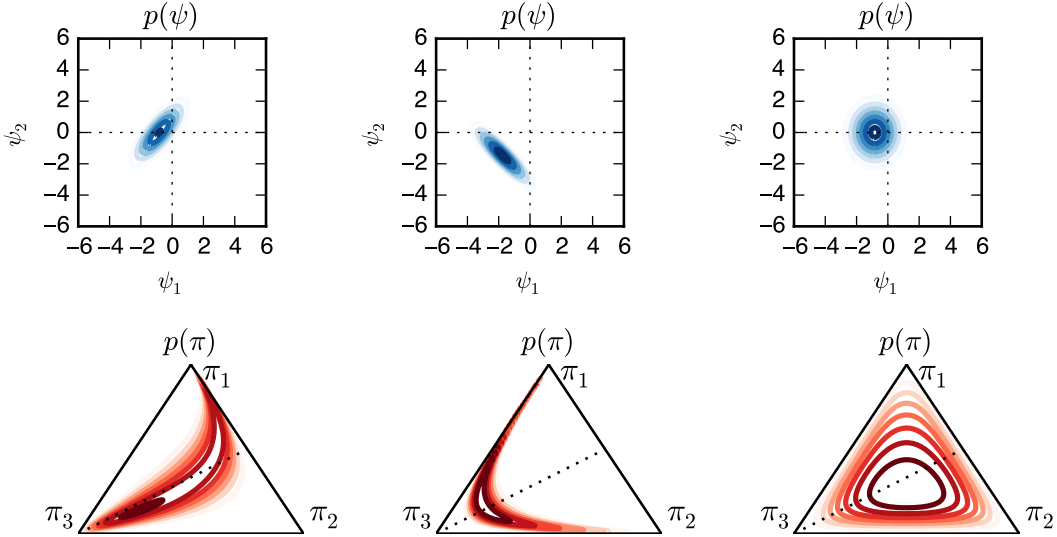

Figure 1: Correlated 2D Gaussian priors on $\boldsymbol{\psi}$ and their implied densities on $\boldsymbol{\pi}_{\text{SB}}(\boldsymbol{\psi})$. See text for details.

where $N_1 = N = \sum_k x_k$ and $\widetilde{\pi}_1 = \pi_1$. For convenience, we define $N(\boldsymbol{x}) \equiv [N_1, \ldots, N_{K-1}]$. This decomposition of the multinomial density is a "stick-breaking" representation where each $\widetilde{\pi}_k$ represents the fraction of the remaining probability mass assigned to the $k$-th component. We let $\widetilde{\pi}_k = \sigma(\psi_k)$, where $\sigma(\cdot)$ denotes the logistic function, and define the function, $\boldsymbol{\pi}_{\text{SB}} : \mathbb{R}^{K-1} \to [0, 1]^K$, which maps a vector $\boldsymbol{\psi}$ to a normalized probability vector $\boldsymbol{\pi}$.

Next, we rewrite the density into the form required by (1) by substituting $\sigma(\psi_k)$ for $\widetilde{\pi}_k$:

$$\text{Mult}(\boldsymbol{x} \mid N, \boldsymbol{\psi}) = \prod_{k=1}^{K-1} \text{Bin}(x_k \mid N_k, \sigma(\psi_k)) = \prod_{k=1}^{K-1} \binom{N_k}{x_k} \sigma(\psi_k)^{x_k}(1 - \sigma(\psi_k))^{N_k - x_k} \quad (7)$$

$$= \prod_{k=1}^{K-1} \binom{N_k}{x_k} \frac{(e^{\psi_k})^{x_k}}{(1 + e^{\psi_k})^{N_k}}. \quad (8)$$

Choosing $a_k(x) = x_k$ and $b_k(x) = N_k$ for each $k = 1, 2, \ldots, K-1$, we can then introduce Pólya-gamma auxiliary variables $\omega_k$ corresponding to each coordinate $\psi_k$; dropping terms that do not depend on $\boldsymbol{\psi}$ and completing the square yields

$$p(\boldsymbol{x}, \boldsymbol{\omega} \mid \boldsymbol{\psi}) \propto \prod_{k=1}^{K-1} e^{(x_k - N_k/2)\psi_k - \omega_k \psi_k^2/2} \propto \mathcal{N}\left(\boldsymbol{\Omega}^{-1}\kappa(\boldsymbol{x}) \,\middle|\, \boldsymbol{\psi}, \boldsymbol{\Omega}^{-1}\right), \quad (9)$$

where $\boldsymbol{\Omega} \equiv \text{diag}(\boldsymbol{\omega})$ and $\kappa(\boldsymbol{x}) \equiv \boldsymbol{x} - N(\boldsymbol{x})/2$. That is, conditioned on $\boldsymbol{\omega}$, the likelihood of $\boldsymbol{\psi}$ under the augmented multinomial model is proportional to a diagonal Gaussian distribution.

Figure 1 shows how several Gaussian densities map to probability densities on the simplex. Correlated Gaussians (left) put most probability mass near the $\pi_1 = \pi_2$ axis of the simplex, and anti-correlated Gaussians (center) put mass along the sides where $\pi_1$ is large when $\pi_2$ is small and vice-versa. Finally, a nearly isotropic Gaussian approximates a symmetric Dirichlet. Appendix A gives a closed-form expression for the density on $\boldsymbol{\pi}$ induced by a Gaussian distribution on $\boldsymbol{\psi}$, and also an expression for a diagonal Gaussian that approximates a Dirichlet by matching moments.

## 3  Correlated topic models

The Latent Dirichlet Allocation (LDA) [8] is a popular model for learning topics from text corpora. The Correlated Topic Model (CTM) [7] extends LDA by including a Gaussian correlation structure among topics. This correlation model is powerful not only because it reveals correlations among

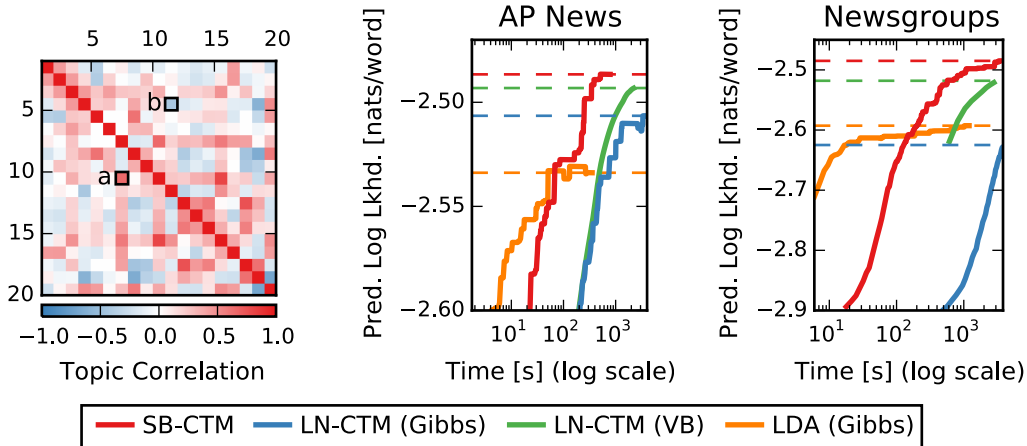

Figure 2: A comparison of correlated topic model performance. The left panel shows a subset of the inferred topic correlations for the AP News corpus. Two examples are highlighted: a) positive correlation between topics (*house, committee, congress, law*) and (*Bush, Dukakis, president, campaign*), and b) anticorrelation between (*percent, year, billion, rate*) and (*court, case, attorney, judge*). The middle and right panels demonstrate the efficacy of our SB-CTM relative to competing models on the AP News corpus and the 20 Newsgroup corpus, respectively.

topics but also because inferring such correlations can significantly improve predictions, especially when inferring the remaining words in a document after only a few have been revealed [7]. However, the addition of this Gaussian correlation structure breaks the Dirichlet-Multinomial conjugacy of LDA, making estimation and particularly Bayesian inference and model-averaged predictions more challenging. An approximate maximum likelihood approach using variational EM [7] is often effective, but a fully Bayesian approach which integrates out parameters may be preferable, especially when making predictions based on a small number of revealed words in a document. A recent Bayesian approach based on a Pólya-Gamma augmentation to the logistic normal CTM (LN-CTM) [5] provides a Gibbs sampling algorithm with conjugate updates, but the Gibbs updates are limited to single-site resampling of one scalar at a time, which can lead to slow mixing in correlated models.

In this section we show that MCMC sampling in a correlated topic model based on the stick breaking construction (SB-CTM) can be significantly more efficient than sampling in the LN-CTM while maintaining the same integration advantage over EM.

In the standard LDA model, each topic $\boldsymbol{\beta}_t$ ($t = 1, 2, \ldots, T$) is a distribution over a vocabulary of $V$ possible words, and each document $d$ has a distribution over topics $\boldsymbol{\theta}_d$ ($d = 1, 2, \ldots, D$). The $n$-th word in document $d$ is denoted $w_{n,d}$ for $d = 1, 2, \ldots, N_d$. When each $\boldsymbol{\beta}_t$ and $\boldsymbol{\theta}_d$ is given a symmetric Dirichlet prior with parameters $\alpha_\beta$ and $\alpha_\theta$, respectively, the generative model is

$$\boldsymbol{\beta}_t \sim \mathrm{Dir}(\alpha_\beta), \quad \boldsymbol{\theta}_d \sim \mathrm{Dir}(\alpha_\theta), \quad z_{n,d} \,|\, \boldsymbol{\theta}_d \sim \mathrm{Cat}(\boldsymbol{\theta}_d), \quad w_{n,d} \,|\, z_{n,d}, \{\boldsymbol{\beta}_t\} \sim \mathrm{Cat}(\boldsymbol{\beta}_{z_{n,d}}). \quad (10)$$

The CTM replaces the Dirichlet prior on each $\boldsymbol{\theta}_d$ with a correlated prior induced by first sampling a correlated Gaussian vector $\boldsymbol{\psi}_d \sim \mathcal{N}(\boldsymbol{\mu}, \boldsymbol{\Sigma})$ and then applying the logistic normal map: $\boldsymbol{\theta}_d = \boldsymbol{\pi}_{\mathsf{LN}}(\boldsymbol{\psi}_d)$ Analogously, our SB-CTM generates the correlation structure by instead applying the stick-breaking logistic map, $\boldsymbol{\theta}_d = \boldsymbol{\pi}_{\mathsf{SB}}(\boldsymbol{\psi}_d)$. The goal is then to infer the posterior distribution over the topics $\boldsymbol{\beta}_t$, the documents' topic allocations $\boldsymbol{\psi}_d$, and their mean and correlation structure $(\boldsymbol{\mu}, \boldsymbol{\Sigma})$, where the parameters $(\boldsymbol{\mu}, \boldsymbol{\Sigma})$ are given a conjugate normal-inverse Wishart (NIW) prior. Modeling correlation structure within the topics $\boldsymbol{\beta}$ can be done analogously.

For fully Bayesian inference in the SB-CTM, we develop a Gibbs sampler that exploits the block conditional Gaussian structure provided by the stick-breaking construction. The Gibbs sampler iteratively samples $\boldsymbol{z} \,|\, \boldsymbol{w}, \boldsymbol{\beta}, \boldsymbol{\psi}$; $\boldsymbol{\beta} \,|\, \boldsymbol{z}, \boldsymbol{w}$; $\boldsymbol{\psi} \,|\, \boldsymbol{z}, \boldsymbol{\mu}, \boldsymbol{\Sigma}, \boldsymbol{\omega}$; and $\boldsymbol{\mu}, \boldsymbol{\Sigma} \,|\, \boldsymbol{\psi}$ as well as the auxiliary variables $\boldsymbol{\omega} \,|\, \boldsymbol{\psi}, \boldsymbol{z}$. The first two are standard updates for LDA models, so we focus on the latter three. Using the identities derived in Section 2.1, the conditional density of each $\boldsymbol{\psi}_d \,|\, \boldsymbol{z}_d, \boldsymbol{\mu}, \boldsymbol{\Sigma}, \boldsymbol{\omega}$ can be written

$$p(\boldsymbol{\psi}_d \,|\, \boldsymbol{z}_d, \boldsymbol{\omega}_d) \; \propto \; \mathcal{N}(\boldsymbol{\Omega}_d^{-1}\kappa(\boldsymbol{c}_d) \,|\, \boldsymbol{\psi}_d, \boldsymbol{\Omega}_d^{-1}) \, \mathcal{N}(\boldsymbol{\psi}_d \,|\, \boldsymbol{\mu}, \boldsymbol{\Sigma}) \; \propto \; \mathcal{N}(\boldsymbol{\psi}_d \,|\, \widetilde{\boldsymbol{\mu}}, \widetilde{\boldsymbol{\Sigma}}), \quad (11)$$

where we have defined

$$\widetilde{\boldsymbol{\mu}} = \widetilde{\boldsymbol{\Sigma}} \left[ \kappa(\boldsymbol{c}_d) + \boldsymbol{\Sigma}^{-1} \boldsymbol{\mu} \right], \quad \widetilde{\boldsymbol{\Sigma}} = \left[ \boldsymbol{\Omega}_d + \boldsymbol{\Sigma}^{-1} \right]^{-1}, \quad c_{d,t} = \sum_n \mathbb{I}[z_{n,d} = t], \quad \boldsymbol{\Omega}_d = \mathrm{diag}(\boldsymbol{\omega}_d),$$

and so it is resampled as a joint Gaussian. The correlation structure parameters $\boldsymbol{\mu}$ and $\boldsymbol{\Sigma}$ are sampled from their conditional NIW distribution. Finally, the auxiliary variables $\boldsymbol{\omega}$ are sampled as Pólya-Gamma random variables, with $\boldsymbol{\omega}_d \,|\, \boldsymbol{z}_d, \boldsymbol{\psi}_d \sim \mathrm{PG}(N(\boldsymbol{c}_d), \boldsymbol{\psi}_d)$. A feature of the stick-breaking construction is that the the auxiliary variable update is embarrassingly parallel.

We compare the performance of this Gibbs sampling algorithm for the SB-CTM to the Gibbs sampling algorithm of the LN-CTM [5], which uses a different Pólya-gamma augmentation, as well as the original variational EM algorithm for the CTM and collapsed Gibbs sampling in standard LDA. Figure 2 shows results on both the AP News dataset and the 20 Newsgroups dataset, where models were trained on a random subset of 95% of the complete documents and tested on the remaining 5% by estimating held-out likelihoods of half the words given the other half. The collapsed Gibbs sampler for LDA is fast but because it does not model correlations its ability to predict is significantly constrained. The variational EM algorithm for the CTM is reasonably fast but its point estimate doesn't quite match the performance from integrating out parameters via MCMC in this setting. The LN-CTM Gibbs sampler continues to improve slowly but is limited by its single-site updates, while the SB-CTM sampler seems to both mix effectively and execute efficiently due to its block Gaussian updating.

The SB-CTM demonstrates that the stick-breaking construction and corresponding Pólya-Gamma augmentation makes inference in correlated topic models both easy to implement and computationally efficient. The block conditional Gaussianity also makes inference algorithms modular and compositional: the construction immediately extends to dynamic topic models (DTMs) [9], in which the latent $\boldsymbol{\psi}_d$ evolve according to linear Gaussian dynamics, and inference can be implemented simply by applying off-the-shelf code for Gaussian linear dynamical systems (see Section 5). Finally, because LDA is so commonly used as a component of other models (e.g. for images [10]), easy, effective, modular inference for CTMs and DTMs is a promising general tool.

## 4 Gaussian processes with multinomial observations

Consider the United States census data, which lists the first names of children born in each state for the years 1910-2013. Suppose we wish to predict the probability of a particular name in New York State in the years 2012 and 2013 given observed names in earlier years. We might reasonably expect that name probabilities vary smoothly over time as names rise and fall in popularity, and that name probability would be similar in neighboring states. A Gaussian process naturally captures these prior intuitions about spatiotemporal correlations, but the observed name counts are most naturally modeled as multinomial draws from latent probability distributions over names for each combination of state and year. We show how efficient inference can be performed in this otherwise difficult model by leveraging the Pólya-gamma augmentation.

Let $\boldsymbol{Z} \in \mathbb{R}^{M \times D}$ denote the matrix of $D$ dimensional inputs and $\boldsymbol{X} \in \mathbb{N}^{M \times K}$ denote the observed $K$ dimensional count vectors for each input. In our example, each row $\boldsymbol{z}_m$ of $\boldsymbol{Z}$ corresponds to the year, latitude, and longitude of an observation, and $K$ is the number of names. Underlying these observations we introduce a set of latent variables, $\psi_{m,k}$ such that the probability vector at input $\boldsymbol{z}_m$ is $\boldsymbol{\pi}_m = \boldsymbol{\pi}_{\mathsf{SB}}(\boldsymbol{\psi}_{m,:})$. The auxiliary variables for the $k$-th name, $\boldsymbol{\psi}_{:,k}$, are linked via a Gaussian process with covariance matrix, $\boldsymbol{C}$, whose entry $C_{i,j}$ is the covariance between input $\boldsymbol{z}_i$ and $\boldsymbol{z}_j$ under the GP prior, and mean vector $\boldsymbol{\mu}_k$. The covariance matrix is shared by all names, and the mean is empirically set to match the measured name probability. The full model is then,

$$\boldsymbol{\psi}_{:,k} \sim \mathcal{GP}(\boldsymbol{\mu}_k, \boldsymbol{C}), \qquad\qquad \boldsymbol{x}_m \sim \mathrm{Mult}(N_m, \boldsymbol{\pi}_{\mathsf{SB}}(\boldsymbol{\psi}_{m,:})).$$

To perform inference, introduce auxiliary Pólya-gamma variables, $\omega_{m,k}$ for each $\psi_{m,k}$. Conditioned on these variables, the conditional distribution of $\boldsymbol{\psi}_{:,k}$ is,

$$p(\boldsymbol{\psi}_{:,k} \,|\, \boldsymbol{Z}, \boldsymbol{X}, \boldsymbol{\omega}, \boldsymbol{\mu}, \boldsymbol{C}) \propto \mathcal{N}\left( \boldsymbol{\Omega}_k^{-1} \kappa(\boldsymbol{X}_{:,k}) \,\Big|\, \boldsymbol{\psi}_{:,k}, \boldsymbol{\Omega}_k^{-1} \right) \mathcal{N}(\boldsymbol{\psi}_{:,k} \,|\, \boldsymbol{\mu}_k, \boldsymbol{C}) \propto \mathcal{N}\left( \boldsymbol{\psi}_{:,k} \,|\, \widetilde{\boldsymbol{\mu}}_k, \widetilde{\boldsymbol{\Sigma}}_k \right)$$

$$\widetilde{\boldsymbol{\Sigma}}_k = \left( \boldsymbol{C}^{-1} + \boldsymbol{\Omega}_k \right)^{-1} \qquad \widetilde{\boldsymbol{\mu}}_k = \widetilde{\boldsymbol{\Sigma}}_k \left( \kappa(\boldsymbol{X}_{:,k}) + \boldsymbol{C}^{-1} \boldsymbol{\mu}_k \right),$$

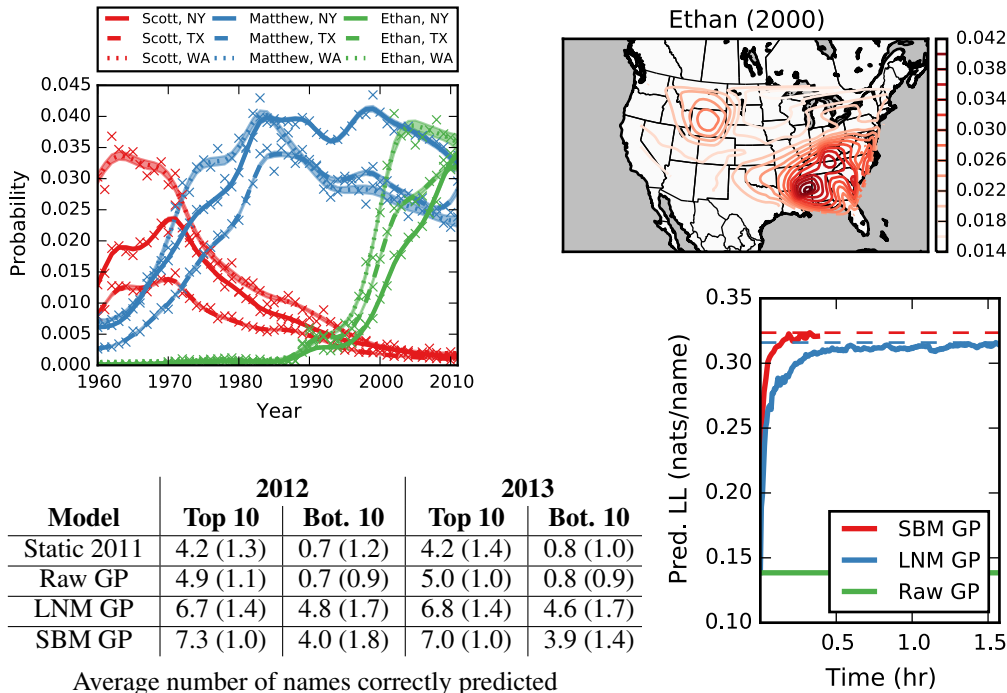

| Model | 2012 | | 2013 | |
|---|---|---|---|---|
| | **Top 10** | **Bot. 10** | **Top 10** | **Bot. 10** |
| Static 2011 | 4.2 (1.3) | 0.7 (1.2) | 4.2 (1.4) | 0.8 (1.0) |
| Raw GP | 4.9 (1.1) | 0.7 (0.9) | 5.0 (1.0) | 0.8 (0.9) |
| LNM GP | 6.7 (1.4) | 4.8 (1.7) | 6.8 (1.4) | 4.6 (1.7) |
| SBM GP | 7.3 (1.0) | 4.0 (1.8) | 7.0 (1.0) | 3.9 (1.4) |

Average number of names correctly predicted

Figure 3: A spatiotemporal Gaussian process applied to the names of children born in the United States from 1960-2013. With a limited dataset of only 50 observations per state/year, the stick breaking and logistic normal multinomial GPs (SBM GP and LNM GP) outperform naïve approaches in predicting the top and bottom 10 names (bottom left, parentheses: std. error). Our SBM GP, which leverages the Pólya-gamma augmentation, is considerably more efficient than the non-conjugate LNM GP (bottom right).

where $\mathbf{\Omega}_k = \mathrm{diag}(\boldsymbol{\omega}_{:,k})$. The auxiliary variables are updated according to their conditional distribution: $\omega_{m,k} \mid \boldsymbol{x}_m, \psi_{m,k} \sim \mathrm{PG}(N_{m,k}, \psi_{m,k})$, where $N_{m,k} = N_m - \sum_{j<k} x_{m,j}$.

Figure 3 illustrates the power of this approach on U.S. census data. The top two plots show the inferred probabilities under our stick-breaking multinomial GP model for the full dataset. Interesting spatiotemporal correlations in name probability are uncovered. In this large-count regime, the posterior uncertainty is negligible since we observe thousands of names per state and year, and simply modeling the transformed empirical probabilities with a GP works well. However, in the sparse data regime with only $N_m = 50$ observations per input, it greatly improves performance to model uncertainty in the latent probabilities using a Gaussian process with multinomial observations.

The bottom panels compare four methods of predicting future names in the years 2012 and 2013 for a down-sampled dataset with $N_m = 50$: predicting based on the empirical probability measured in 2011; a standard GP to the empirical probabilities transformed by $\boldsymbol{\pi}_{\mathsf{SB}}^{-1}$ (Raw GP); a GP whose outputs are transformed by the logistic normal function, $\boldsymbol{\pi}_{\mathsf{LN}}$, to obtain multinomial probabilities (LNM GP) fit using elliptical slice sampling [2]; and our stick-breaking multinomial GP (SBM GP). In terms of ability to predict the top and bottom 10 names, the multinomial models are both comparable and vastly superior to the naive approaches.

The SBM GP model is considerably faster than the logistic normal version, as shown in the bottom right panel. The augmented Gibbs sampler is more efficient than the elliptical slice sampling algorithm used to handle the non-conjugacy in the LNM GP. Moreover, we are able to make collapsed predictions in which we compute the predictive distribution test $\psi$'s given $\omega$, integrating out the training $\psi$. In contrast, the LNM GP must condition on the training GP values in order to make predictions, and effectively integrate over training samples using MCMC. Appendix B goes into greater detail on how marginal predictions are computed and why they are more efficient than predicting conditioned on a single value of $\psi$.

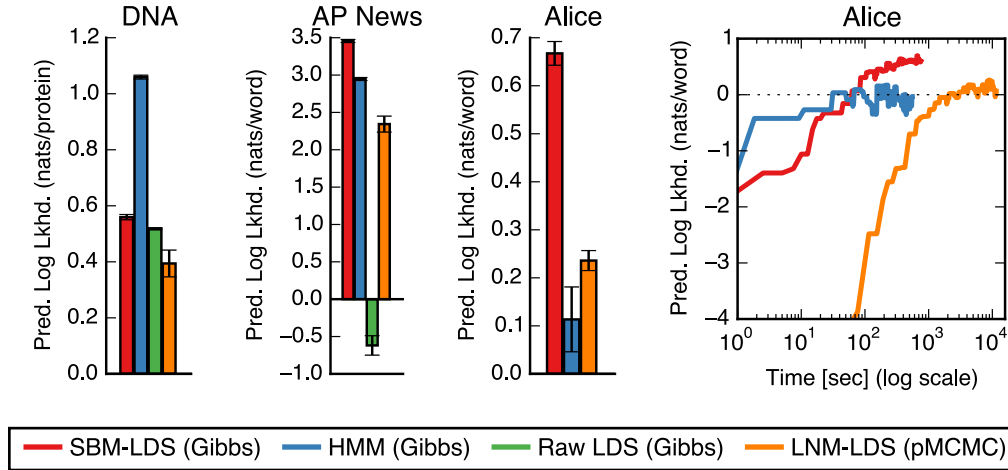

Figure 4: Predictive log likelihood comparison of time series models with multinomial observations.

# 5 Multinomial linear dynamical systems

While discrete-state hidden Markov models (HMMs) are ubiquitous for modeling time series and sequence data, it can be preferable to use a continuous state space model. In particular, while discrete states have no intrinsic geometry, continuous states can correspond to natural Euclidean embeddings [11]. These considerations are particularly relevant to text, where word embeddings [12] have proven to be a powerful tool.

Gaussian linear dynamical systems (LDS) provide very efficient learning and inference algorithms, but they can typically only be applied when the observations are themselves linear with Gaussian noise. While it is possible to apply a Gaussian LDS to count vectors [11], the resulting model is misspecified in the sense that, as a continuous density, the model assigns zero probability to training and test data. However, Belanger and Kakade [11] show that this model can still be used for several machine learning tasks with compelling performance, and that the efficient algorithms afforded by the misspecified Gaussian assumptions confer a significant computational advantage. Indeed, the authors have observed that such a Gaussian model is "worth exploring, since multinomial models with softmax link functions prevent closed-form M step updates and require expensive" computations [13]; this paper aims to bridge precisely this gap and enable efficient Gaussian LDS computational methods to be applied while maintaining multinomial emissions and an asymptotically unbiased representation of the posterior. While there are other approximation schemes that effectively extend some of the benefits of LDSs to nonlinear, non-Gaussian settings, such as the extended Kalman filter (EKF) and unscented Kalman filter (UKF) [14, 15], these methods do not allow for asymptotically unbiased Bayesian inference, can have complex behavior, and can make model learning a challenge. Alternatively, particle MCMC (pMCMC) [1] is a very powerful algorithm that provides unbiased Bayesian inference for very general state space models, but it does not enjoy the efficient block updates or conjugacy of LDSs or HMMs.

The stick-breaking multinomial linear dynamical system (SBM-LDS) generates states via a linear Gaussian dynamical system but generates multinomial observations via the stick-breaking map:

$$\boldsymbol{z}_0|\boldsymbol{\mu}_0, \boldsymbol{\Sigma}_0 \sim \mathcal{N}(\boldsymbol{\mu}_0, \boldsymbol{\Sigma}_0), \quad \boldsymbol{z}_t|\boldsymbol{z}_{t-1}, \boldsymbol{A}, \boldsymbol{B} \sim \mathcal{N}(\boldsymbol{A}\boldsymbol{z}_{t-1}, \boldsymbol{B}), \quad \boldsymbol{x}_t|\boldsymbol{z}_t, \boldsymbol{C} \sim \mathrm{Mult}(N_t, \boldsymbol{\pi}_{\mathsf{SB}}(\boldsymbol{C}\boldsymbol{z}_t)),$$

where $\boldsymbol{z}_t \in \mathbb{R}^D$ is the system state at time $t$ and $\boldsymbol{x}_t \in \mathbb{N}^K$ are the multinomial observations. We suppress notation for conditioning on $\boldsymbol{A}$, $\boldsymbol{B}$, $\boldsymbol{C}$, $\boldsymbol{\mu}_0$, and $\boldsymbol{\Sigma}_0$, which are system parameters of appropriate sizes that are given conjugate priors. The logistic normal multinomial LDS (LNM-LDS) is defined analogously but uses $\boldsymbol{\pi}_{\mathsf{LN}}$ in place of $\boldsymbol{\pi}_{\mathsf{SB}}$.

To produce a Gibbs sampler with fully conjugate updates, we augment the observations with Pólya-gamma random variables $\omega_{t,k}$. As a result, the conditional state sequence $\boldsymbol{z}_{1:T}|\boldsymbol{\omega}_{1:T}, \boldsymbol{x}_{1:T}$ is jointly distributed according to a Gaussian LDS in which the diagonal observation potential at time $t$ is $\mathcal{N}(\boldsymbol{\Omega}_t^{-1}\kappa(\boldsymbol{x}_t)|\boldsymbol{C}\boldsymbol{z}_t, \boldsymbol{\Omega}_t^{-1})$. Thus the state sequence can be jointly sampled using off-

the-shelf LDS software, and the system parameters can similarly be updated using standard algorithms. The only remaining update is to the auxiliary variables, which are sampled according to $\boldsymbol{\omega}_t | \boldsymbol{z}_t, \boldsymbol{C}, \boldsymbol{x} \sim \mathrm{PG}(N(\boldsymbol{x}_t), \boldsymbol{C}\boldsymbol{z}_t)$.

We compare the SBM-LDS and the Gibbs sampling inference algorithm to three baseline methods: an LNM-LDS using pMCMC and ancestor resampling [16] for inference, an HMM using Gibbs sampling, and a "raw" LDS which treats the multinomial observation vectors as observations in $\mathbb{R}^K$ as in [11]. We examine each method's performance on each of three experiments: in modeling a sequence of 682 amino acids from human DNA with 22 dimensional observations, a set of 20 random AP news articles with an average of 77 words per article and a vocabulary size of 200 words, and an excerpt of 4000 words from Lewis Carroll's *Alice's Adventures in Wonderland* with a vocabulary of 1000 words. We reserved the final 10 amino acids, 10 words per news article, and 100 words from *Alice* for computing predictive likelihoods. Each linear dynamical model had a 10-dimensional state space, while the HMM had 10 discrete states (HMMs with 20, 30, and 40 states all performed worse on these tasks).

Figure 4 (left panels) shows the predictive log likelihood for each method on each experiment, normalized by the number of counts in the test dataset and relative to the likelihood under a multinomial model fit to the training data mean. For the DNA data, which has the smallest "vocabulary" size, the HMM achieves the highest predictive likelihood, but the SBM-LDS edges out the other LDS methods. On the two text datasets, the SBM-LDS outperforms the other methods, particularly in *Alice* where the vocabulary is larger and the document is longer. In terms of run time, the SBM-LDS is orders of magnitude faster than the LNM-LDS with pMCMC (right panel) because it mixes much more efficiently over the latent trajectories.

# 6 Related Work

The stick-breaking transformation used herein was applied to categorical models by Khan et al. [17], but they used local variational bound instead of the Pólya-gamma augmentation. Their promising results corroborate our findings of improved performance using this transformation. Their generalized expectation-maximization algorithm is not fully Bayesian, and does not integrate into existing Gaussian modeling and inference code as easily as our augmentation.

Conversely, Chen et al. [5] used the Pólya-gamma augmentation in conjunction with the logistic normal transformation for correlated topic modeling, exploiting the conditional conjugacy of a single entry $\psi_k | \omega_k, \boldsymbol{\psi}_{\neg k}$ with a Gaussian prior. Unlike our stick-breaking transformation, which admits block Gibbs sampling over the entire vector $\boldsymbol{\psi}$ simultaneously, their approach is limited to single-site Gibbs sampling. As shown in our correlated topic model experiments, this has dramatic effects on inferential performance. Moreover, it precludes analytical marginalization and integration with existing Gaussian modeling algorithms. For example, it is not immediately applicable to inference in linear dynamical systems with multinomial observations.

# 7 Conclusion

These case studies demonstrate that the stick-breaking multinomial model construction paired with the Pólya-gamma augmentation yields a flexible class of models with easy, efficient, and compositional inference. In addition to making these models easy, the methods developed here can also enable new models for multinomial and mixed data: the latent continuous structures used here to model correlations and state-space structure can be leveraged to explore new models for interpretable feature embeddings, interacting time series, and dependence with other covariates.

# 8 Acknowledgements

S.W.L. is supported by a Siebel Scholarship and the Center for Brains, Minds and Machines (CBMM), funded by NSF STC award CCF-1231216. M.J.J. is supported by the Harvard/MIT Joint Research Grants Program. R.P.A. is supported by NSF IIS-1421780 as well as the Alfred P. Sloan Foundation.

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
