[Supplementary Material · appendix.pdf]

# Dependent Multinomial Models Made Easy: Supplementary Material

**Scott W. Linderman**[*]
Harvard University
Cambridge, MA 02138
swl@seas.harvard.edu

**Matthew J. Johnson**[*]
Harvard University
Cambridge, MA 02138
mattjj@csail.mit.edu

**Ryan P. Adams**
Twitter & Harvard University
Cambridge, MA 02138
rpa@seas.harvard.edu

## A    Symmetry breaking in $\boldsymbol{\pi}_{\mathsf{SB}}$

The stick-breaking map $\boldsymbol{\pi}_{\mathsf{SB}} : \mathbb{R}^K \to [0,1]^K$ is asymmetric in the sense that while the logistic map $\boldsymbol{\pi}_{\mathsf{LN}} : \mathbb{R}^K \to [0,1]^K$ can be written as the composition of an coordinate-wise logistic function and a normalization,

$$\boldsymbol{\pi}_{\mathsf{LN}}(\psi) = \left( \boldsymbol{\pi}_{\mathsf{LN}}^{(1)}(\psi), \quad \cdots, \quad \boldsymbol{\pi}_{\mathsf{LN}}^{(K)}(\psi) \right) \qquad \boldsymbol{\pi}_{\mathsf{LN}}^{(k)}(\psi) = \frac{e^{\psi_k}}{\sum_{j=1}^K e^{\psi_j}}, \tag{1}$$

the stick-breaking map does not have such a coordinate-wise separation:

$$\boldsymbol{\pi}_{\mathsf{SB}}(\psi) = \left( \boldsymbol{\pi}_{\mathsf{LN}}^{(1)}(\psi), \quad \cdots, \quad \boldsymbol{\pi}_{\mathsf{SB}}^{(K)}(\psi) \right) \qquad \boldsymbol{\pi}_{\mathsf{SB}}^{(k)}(\psi) = \sigma(\psi_k) \left( \sum_{j<k} \sigma(\psi_j) \right). \tag{2}$$

In particular, $\boldsymbol{\pi}_{\mathsf{SB}}$ does not preserve permutation symmetries in the density $p(\psi)$, so that while for any permutation matrix $P$ we have

$$p(P\psi) = p(\psi) \implies p(P\boldsymbol{\pi}_{\mathsf{LN}}(\psi)) = p(\boldsymbol{\pi}_{\mathsf{LN}}(\psi)) \tag{3}$$

the same does not hold when $\boldsymbol{\pi}_{\mathsf{LN}}$ is replaced with $\boldsymbol{\pi}_{\mathsf{SB}}$. As a result, the stick-breaking model used in this paper (and in Khan et al. [1]) yields priors (and posteriors) that are not invariant to relabeling of the entries of the corresponding multinomial parameter or the multinomial counts themselves. See Figure 1 and compare it to Figure 1 of the main text.

This symmetry breaking may be undesirable in some cases, but in the models we have studied so far (and in those studied in Khan et al. [1]) the effect does not seem detrimental in terms of learning informative correlation structures or in terms of model predictions. For example, in the correlated topic model (CTM) studied in Section 3, the model is unidentifiable up to permutation on the topic labels and therefore breaking this symmetry does not reduce its representational capacity. For models in which the counts from multinomials with correlated parameters are observed directly, such as in the models of Sections 4 and 5, based on the experiments in this paper the loss of symmetry does not seem to impact performance while the inference advantages are significant. See also the discussion in Khan et al. [1, Section 3].

## B    Transforming between $p(\boldsymbol{\psi})$ and $p(\boldsymbol{\pi})$

Since the mapping between $\boldsymbol{\pi}$ and $\boldsymbol{\psi}$ is invertible, we can compute the distribution on $\boldsymbol{\pi}$ that is implied by a Gaussian distribution on $\boldsymbol{\psi}$. Assume $\boldsymbol{\psi} \sim \mathcal{N}(\boldsymbol{\mu}, \boldsymbol{\Sigma})$. Then,

$$p(\boldsymbol{\pi} \,|\, \boldsymbol{\mu}, \boldsymbol{\Sigma}) = \mathcal{N}(\boldsymbol{\pi}_{\mathsf{SB}}^{-1}(\boldsymbol{\pi}) \,|\, \boldsymbol{\mu}, \boldsymbol{\Sigma}) \left| \frac{\mathrm{d}\boldsymbol{\psi}}{\mathrm{d}\boldsymbol{\pi}} \right|$$

---

[*]These authors contributed equally.

Figure 1: Correlated 2D Gaussian priors on $\psi$ and their implied densities on $\pi_{\text{LN}}(\psi)$. Compare to Figure 1 of the main text, which shows an analogous plot of implied densities on $\pi_{\text{SB}}(\psi)$.

From above, we have

$$
\psi_1 = \sigma^{-1}(\pi_1), \qquad \psi_2 = \sigma^{-1}\left(\frac{\pi_2}{1-\pi_1}\right), \qquad \cdots, \qquad \psi_k = \sigma^{-1}\left(\frac{\pi_k}{1-\sum_{j<k}\pi_j}\right).
$$

Let

$$
g(x) = \left.\frac{\mathrm{d}\sigma^{-1}(x)}{\mathrm{d}x}\right|_{x=x} = \frac{\mathrm{d}}{\mathrm{d}x}\log\left(\frac{x}{1-x}\right) = \frac{1}{x} + \frac{1}{1-x} = \frac{1}{x(1-x)}.
$$

Then,

$$
\frac{\partial\psi_1}{\partial\pi_1} = g(\pi_1), \qquad \frac{\partial\psi_k}{\partial\pi_k} = g\left(\frac{\pi_k}{1-\sum_{j<k}\pi_j}\right)\frac{1}{1-\sum_{j<k}\pi_j}, \qquad \frac{\partial\psi_k}{\partial\pi_{j>k}} = 0.
$$

Since the Jacobian of the inverse transformation is lower triangular, its determinant is simply the product of its diagonal entries,

$$
\left|\frac{\mathrm{d}\psi}{\mathrm{d}\pi}\right| = \prod_{k=1}^{K}\left[g\left(\frac{\pi_k}{1-\sum_{j<k}\pi_j}\right)\frac{1}{1-\sum_{j<k}\pi_j}\right]
$$

$$
= \prod_{k=1}^{K}\left[\frac{1-\sum_{j<k}\pi_j}{\pi_k}\frac{1-\sum_{j<k}\pi_j}{1-\sum_{j<k}\pi_j-\pi_k}\frac{1}{1-\sum_{j<k}\pi_j}\right]
$$

$$
= \prod_{k=1}^{K}\left[\frac{1-\sum_{j=1}^{k-1}\pi_j}{\pi_k(1-\sum_{j=1}^{k}\pi_j)}\right]
$$

Thus, the final density is,

$$
p(\pi\,|\,\mu,\Sigma) = \mathcal{N}(\pi_{\text{SB}}^{-1}(\pi)\,|\,\mu,\Sigma)\cdot\prod_{k=1}^{K}\left[\frac{1-\sum_{j=1}^{k-1}\pi_j}{\pi_k(1-\sum_{j=1}^{k}\pi_j)}\right].
$$

Now, suppose we are given a Dirichlet distribution, $\pi \sim \text{Dir}(\pi\,|\,\alpha)$, and we wish to compute the density on $\psi$. We have,

$$
p(\psi\,|\,\alpha) = \text{Dir}(\pi_{\text{SB}}(\psi)\,|\,\alpha)\cdot\left|\frac{\mathrm{d}\pi}{\mathrm{d}\psi}\right|
$$

$$
= \text{Dir}(\pi_{\text{SB}}(\psi)\,|\,\alpha)\cdot\prod_{k=1}^{K}\left[\frac{\pi_k(1-\sum_{j=1}^{k}\pi_j)}{1-\sum_{j=1}^{k-1}\pi_j}\right],
$$

Figure 2: Density and log density of $p(\boldsymbol{\psi} \mid \boldsymbol{\alpha} = \mathbf{1})$, the density on $\boldsymbol{\psi}$ implied by a $K = 9$ dimensional symmetric Dirichlet density on $\boldsymbol{\pi}$ with parameter $\alpha = 1$.

where we have used the fact that the Jacobian of the inverse transformation is simply the inverse of the Jacobian of the forward transformation. We simply need to rewrite the Jacobian in terms of $\psi$ rather than $\pi$. Note that $1 - \sum_{j<k} \pi_j$ is the length of the remaining stick and $\sigma(\psi_k)$ is the fraction of the remaining "stick" allocated to $\pi_k$. Thus, the remaining stick length is equal to,

$$1 - \sum_{j<k} \pi_j \equiv \prod_{j<k} (1 - \sigma(\psi_j)) \equiv \prod_{j<k} \sigma(-\psi_j).$$

Moreover, $\pi_k = \sigma(\psi_k)(1 - \sum_{j<k} \pi_j) = \sigma(\psi_k) \prod_{j<k} \sigma(-\psi_j)$. Thus,

$$p(\boldsymbol{\psi} \mid \boldsymbol{\alpha}) = \text{Dir}(\boldsymbol{\pi}_{\text{SB}}(\boldsymbol{\psi}) \mid \boldsymbol{\alpha}) \cdot \prod_{k=1}^{K} \left[ \frac{\left( \sigma(\psi_k) \prod_{j<k} \sigma(-\psi_j) \right) \left( \prod_{j \leq k} \sigma(-\psi_j) \right)}{\prod_{j<k} \sigma(-\psi_j)} \right],$$

$$= \text{Dir}(\boldsymbol{\pi}_{\text{SB}}(\boldsymbol{\psi}) \mid \boldsymbol{\alpha}) \cdot \prod_{k=1}^{K} \left[ \sigma(\psi_k) \prod_{j \leq k} \sigma(-\psi_j) \right],$$

Expanding the Dirichlet distribution and substituting $\psi$ for $\pi$, we conclude that,

$$p(\boldsymbol{\psi} \mid \boldsymbol{\alpha}) = \frac{1}{B(\boldsymbol{\alpha})} \prod_{k=1}^{K-1} \sigma(\psi_k)^{\alpha_k} \cdot \sigma(-\psi_k)^{\sum_{j=k+1}^{K} \alpha_j}.$$

This factorized form is unsurprising given that the Dirichlet distribution can be written as a stick-breaking product of beta distributions in the same way that the multinomial can be written as a product of binomials. Each term in the product above corresponds to the transformed beta distribution over $\widetilde{\pi}_k$.

Figure 2 shows the marginal densities on $\psi_k$ implied by a $K = 9$ dimensional symmetric Dirichlet prior on $\boldsymbol{\pi}$ with $\alpha = 1$. The densities of $\psi_k$ become increasingly skewed for small values of $k$, but they are still well approximate by a Gaussian distribution. In order to approximate a uniform distribution, we numerically compute the mean and variance of these densities to set the parameters of a diagonal Guassian distribution.

Figure 3: Marginal density, $p(\boldsymbol{\psi} \mid \boldsymbol{x})$ in red shading along with the ellipses of multivariate normal conditional distribution $p(\boldsymbol{\psi} \mid \boldsymbol{x}, \boldsymbol{\omega})$ for 4 steps of the Gibbs sampler. In Gaussian models where we aim to predict $\boldsymbol{\psi}_{\text{test}}$ on test data, there are substantial gains to be had from making marginal predictions of $\boldsymbol{\psi}_{\text{test}} \mid \boldsymbol{x}, \boldsymbol{\omega}$, integrating out $\boldsymbol{\psi}_{\text{train}}$. The key is that the conditional densities overlap substantially with the marginal density.

## C   Marginal Predictions with the Augmented Model

One of the primary advantages offered by the Pólya-gamma augmentation is the ability to make marginal predictions about $\boldsymbol{\psi}_{\text{test}} \mid \boldsymbol{x}, \boldsymbol{\omega}$, integrating out the value of $\boldsymbol{\psi}_{\text{train}}$. For example, in the GP multinomial regression models described in the main text, the methods were evaluated on the accuracy of their predictions about future name probabilities, which were functions of $\boldsymbol{\psi}_{\text{test}}$. When $p(\boldsymbol{\psi}_{\text{train}})$ and $p(\boldsymbol{\psi}_{\text{test}} \mid \boldsymbol{\psi}_{\text{train}})$ are both Gaussian, we can integrate out the latent training variables in order to predict their test values. In a latent Gaussian-multinomial model, the posterior distribution over those latent training variables is non-Gaussian, but after Pólya-gamma augmentation, it is rendered Gaussian.

With the augmentation, we can write

$$p(\boldsymbol{\psi}_{\text{test}} \mid \boldsymbol{x}) \approx \frac{1}{M} \sum_{m=1}^{M} \int p(\boldsymbol{\psi}_{\text{test}} \mid \boldsymbol{\psi}_{\text{train}}) \, p(\boldsymbol{\psi}_{\text{train}} \mid \boldsymbol{x}, \boldsymbol{\omega}^{(m)}) \, d\boldsymbol{\psi}_{\text{train}} \qquad \boldsymbol{\omega}^{(m)} \sim p(\boldsymbol{\omega} \mid \boldsymbol{x}),$$

and perform Monte Carlo integration over $\boldsymbol{\omega}$ in order to compute the predictive distribution. By contrast, in the standard formulation we must perform Monte Carlo integration over $\boldsymbol{\psi}$,

$$p(\boldsymbol{\psi}_{\text{test}} \mid \boldsymbol{x}) = \frac{1}{M} \sum_{m=1}^{M} p(\boldsymbol{\psi}_{\text{test}} \mid \boldsymbol{\psi}_{\text{train}}^{(m)}) \qquad \boldsymbol{\psi}_{\text{train}}^{(m)} \sim p(\boldsymbol{\psi}_{\text{train}} \mid \boldsymbol{x}).$$

Why does the augmented model confer a predictive advantage? It does not come from performing Monte Carlo integration over a smaller dimension since $\boldsymbol{\omega}$ and $\boldsymbol{\psi}_{\text{train}}$ are of the same size. Instead, it comes from the ability of the conjugate Gibbs sampler to efficiently mix over $\boldsymbol{\psi}$ and $\boldsymbol{\omega}$, and from the ability of a single sample of $\boldsymbol{\omega}$ to render a conditional Gaussian distribution over $\boldsymbol{\psi}$ that captures much of the volume of the true marginal distribution.

This latter point is illustrated in Figure 3. The red shading shows the true marginal density of $\boldsymbol{\psi}$ and the blue ellipses show the conditional density for a fixed value of $\boldsymbol{\omega}$. Each ellipse capture a significant amount of the marginal distribution, indicating that with a single sample of $\boldsymbol{\omega}$ we can integrate over a substantial amount of the uncertainty in $\boldsymbol{\psi}$. This example is only for a $K = 3$ dimensional multinomial observation, but this intuition should extend to higher dimensions in which the advantages of analytical integration should be more readily apparent.

# References

[1] Mohammad E Khan, Shakir Mohamed, Benjamin M Marlin, and Kevin P Murphy. A stick-breaking likelihood for categorical data analysis with latent Gaussian models. In *International Conference on Artificial Intelligence and Statistics*, pages 610–618, 2012.