[Reviews · NeurIPS 2015]

Submitted by Assigned_Reviewer_1

Summary: The paper presents a data-augmentation scheme for performing closed-form Gibbs sampling in models with Gaussian priors and multinomial likelihoods. The proposed scheme is based on introducing Polya-gamma random variables (one per observation) which allows Gaussianifying the multinomial likelihood and leading to closed-form Gibbs sampling updates. The proposed scheme is evaluated on a number of models with Gaussian priors and multinomial likelihood and is shown to demonstrate superior performance when compared to several baselines.

Comments:

Models with Gaussian priors and multinomial likelihoods are known to be troublesome when doing inference due to non-conjugacy. This paper presents a neat trick to "Gaussianify" the multinomial likelihood using an auxiliary variable construction based on the recently proposed Polya-gamma augmentation [3] by Polson and Scott (2012). This allows closed-form Gibbs sampling

The idea of using the Polya-gamma trick has been used in several papers recently for binary data (using logistic link) and count data (using negative binomial link). However, the muiltinomial case hasn't been explored much. I should also mention that the case of multinomial distribution was also considered in the original Polya-gamma paper by Polson and Scott (please see the appendix S6.3: http://arxiv.org/pdf/1205.0310v3.pdf) and used recently in [8]. This aspect should be discussed well before the proposed method is described.

That said, this paper takes a somewhat different approach for handling the multinomial likelihood than the one presented in the appendix of [3]. Although the resulting equations here appear very similar to those obtained in [3], the proposed approach in this paper is more elegant in its construction (using a stick-breaking construction). In particular, this also enables block sampling whereas [8] which used the trick presented in [3] requires component-wise sampling. The experimental results corroborate the advantage of doing the block sampling. This makes the proposed construction very appealing.

I also like the range of applications considered to demonstrate the usefulness and advantage of the proposed inference method over the alternatives.

A minor point: I would be curious to see how much of a computational overhead does sampling the Polya-gamma variables introduces. At least, some discussion of this will be nice to have.

Overall, although built on some standard pieces (Polya-gamma sampling, stick-breaking construction), the proposed inference method provides an easy way to do Gibbs sampling in non-conjugate models and is expected to be useful for a wide range of non-conjugate models with Gaussian priors and multinomial likelihood (as shown in some of the models presented in the paper).
Summary: The paper presents a new algorithm for doing conjugate, Gibbs sampling based inference in Baysian models with multinomial likelihoods (with correlated draws), and demonstrates it via 3 models (correlated topic models, multinomial Gaussian processes, and multinomial linear dynamical systems). The algorithm leverages the recent work on polya-gamma augmentation which allows Gaussianifying likelihoods that are in the form of a logit function. The idea is nicely presented, and compellingly demonstrated via experiments on several models.

Submitted by Assigned_Reviewer_2

The paper utilizes the Polya-Gamma augmentation to model dependent multinomial observations, This building block is demonstrated in 3 use cases: correlated topic modeling, correlated multinomial data with GP latent variables, and LDS with multinomial emissions.

Quality: Using Polya-Gamma augmentation, this paper derives efficient inference algorithms for multiple models involving correlated multinomial observations. The model setup is relatively novel, and from the experimental results, the proposed building block works outperforms baselines consistently in terms of complexity and accuracy.

Clarity: Most part of the paper is clearly written. However, some of the notations are reused, and bold/no-bold notations are not carefully checked. It is suggested that the authors improve on the notations to increase readability.

Originality: The pieces in the work have been proposed in previous work, but the usage of Poly-Gamma augmentation in multinomial data is relatively novel. A few related works are properly cited and acknowledged.

Significance: Though incremental, from the 3 use cases, the building block provided is effective with low computation cost. I wonder whether the software can be made available, which will be helpful for the community.
Summary: The paper applied the recently developed Polya-Gamma augmentation to tackle dependent multinomial models. Though incremental, it provides an effective and efficient approach/building block to model correlated multinomial data. I recommend it as an accept.

Submitted by Assigned_Reviewer_3

The motivation here is EXACTLY right --being able to use the wide variety of gaussian modeling machinery when working with multinomial data -- which has been unfortunately underemphasized in previous work on logistic normal modeling.

I could see the method here being widely used if it works as well as advertised (this was a light review for me so I did not investigate in detail).

Thanks for the comments comparing to Chien et al.

Some other approaches to logistic normal inference that need to be discussed:

laplace approximation http://www.stat.washington.edu/people/pdhoff/Preprints/nhed.ps

other gibbs sampling http://dirichlet.net/pdf/mimno08gibbs.pdf

Also - less directly relevant, but some examples of applications using gaussian priors on top of binomials/multinomials, that might benefit from this overall approach:

http://structuraltopicmodel.com/ http://journals.plos.org/plosone/article?id=10.1371/journal.pone.0113114 http://www.icml-2011.org/papers/534_icmlpaper.pdf http://brenocon.com/oconnor+stewart+smith.irevents.acl2013.pdf

===== update 2016-01-24, before review release: the review above is the same I wrote when reviewing the paper for the first time. I am happy this paper was accepted. However, I will take this opportunity to note that the authors did not address the previous work I hoped they would discuss -- in particular, how their method compares to Hoff 2003, or Mimno et al. 2008 (http://www.stat.washington.edu/people/pdhoff/Preprints/nhed.ps and http://dirichlet.net/pdf/mimno08gibbs.pdf).
Summary: This is a great problem with exactly the right motivation and excellent writing.

I do have some citations these authors missed, but would not be surprised if the method outlined here is superior.

Author Feedback
Author rebuttal: Thanks to all the reviewers for their input. We focus on the critical feedback here.

# Reviewer 2 #

We appreciate that the reviewer finds the ideas to be elegant, nicely presented, and compellingly demonstrated in the experiments.

We agree that the paper can be improved by discussing Appendix S6.3 of [3] directly and early in the paper. Specifically, in addition to some other edits, we will add a sentence to the introduction: "Previous work on applying the PĆ³lya-gamma augmentation to the multinomial [3, 8] only provides single-site Gibbs updating of the multinomial parameter, while this paper develops a complete augmentation in the sense that, given the auxiliary variables, the entire multinomial parameter is resampled as a block in a single Gibbs update." In addition to highlighting the block updating, we will also make the point that for a fixed sample of the auxiliary variables our method allows marginalizing out the multinomial parameter when making predictions, while using the augmentation in [3, 8] requires that marginalization to be done via single-site MCMC.

We also wish to highlight that our experiments in Section 3 compare directly to the single-site augmentation updates. That is, LN-CTM (Gibbs) is the method of [8] using the augmentation from [3, S6.3] restricted to correlated topic models, and our Figure 2 demonstrates that our new augmentation provides improvements by orders of magnitude.

We can also comment on the computational overhead of PG sampling. Briefly, while more expensive than, say, the multinomial, Dirichlet, or NIW steps of the CTM Gibbs sampler, the updates are embarrassingly parallelizable and hence not a significant performance hit, as shown in the experiments.

# Reviewer 4 #

We have improved the notation and fixed errors in bolding.

The software is indeed online, including code to reproduce each experiment as well as library functions for using the stick-breaking construction and, building on existing code to generate PG samples, a multithreaded PG sampler with Python bindings. After the review period, we hope you check it out; it really is "made easy"!

# Reviewer 6 #

Thanks for the references!